# Multi-Index Geophysical Monitoring and Early Warning for Rockburst in Coalmine: A Case Study

**DOI:** 10.3390/ijerph20010392

**Published:** 2022-12-26

**Authors:** Xiaofei Liu, Siqing Zhang, Enyuan Wang, Zhibo Zhang, Yong Wang, Shengli Yang

**Affiliations:** 1School of Safety Engineering, China University of Mining and Technology, Xuzhou 221116, China; 2Key Laboratory of Gas and Fire Control for Coal Mines (China University of Mining and Technology), Ministry of Education, Xuzhou 221116, China; 3Guizhou Anhe Yongzhu Technology Co., Ltd., Guiyang 550000, China; 4School of Civil and Resource Engineering, University of Science and Technology Beijing, Beijing 100083, China; 5China Energy Engineering Group, Zhe Jiang Thermal Power Construction Co., Ltd., Hangzhou 310000, China

**Keywords:** rockburst, microseismic, acoustic emission, electromagnetic radiation, early warning

## Abstract

Rockburst is a major disaster in deep mining, restricting the safety and the production efficiency of the Laohutai Coal Mine in Fushun, Liaoning Province. To predict and prevent coalmine rockbursts, a comprehensive method based on multi-instrument monitoring is proposed by using a YDD16 acoustic-electromagnetic monitor and microseismic monitoring system, including microseismic (MS) monitoring, electromagnetic radiation (EMR) monitoring, and acoustic emission (AE) monitoring. Field investigation shows that MS, AE, and EMR signals have abnormal precursors before rockbursts in a new working face. Based on the fluctuation theory and D-S evidence theory, the multi-index geophysical monitoring and early warning technology for rockburst disasters in the Laohutai Coal Mine are established. The method has been applied to the prediction of rockbursts in the Laohutai Coal Mine. The application shows that the acoustic-electromagnetic synchronous monitoring and early warning technology can accurately identify the potential rockburst risk and trigger an early warning, which is more reliable than a single method. The case study of the Laohutai rockburst shows that the joint early warning method of multi-instrument comprehensive monitoring can predict the possibility of rockbursts.

## 1. Introduction

Rockbursts can cause surrounding rocks to suddenly move out, eject, damage, or block roadways, resulting in casualties and damage to the ventilation system [1,2,3,4]. At present, China is the most prominent country in the world in terms of rockburst disasters, causing hundreds of casualties and huge economic losses in China every year. Relevant statistics show that the number of coalmines with impact ground pressure in China has reached as high as 329, of which 253 are still being mined. As of 2018, over 300 coalmines and 20 non-coalmines in China have experienced impact ground pressure. With the coal mining depths increasing at a rate of 8 m to 12 m per year, especially in the eastern mines, the increase rate even reaches 10 m to 25 m per year. Most of the former state-owned key coalmines have entered deep mining (some have reached a mining depth of more than 1000 m). As a result, underground coalmine stresses continue to increase, stope structures are becoming more and more complex, and the frequency, intensity, and damage of rockbursts have increased significantly.

During the incubation and evolution of rockbursts, coal and rock masses will deform, fracture, or rupture under the superposition of mining stresses and tectonic stresses, and generate signals such as electromagnetic radiation, ultra-low frequency electromagnetic induction, ultrasound, acoustic emission, or infrasound. The changes in these acoustic and electrical signals can effectively reflect the evolution mechanism of rockbursts. Scholars have carried out corresponding studies on the changes in these acoustic and electrical signals in coal and rock masses.

MS [5,6,7], EMR [8,9,10,11], and AE [12,13,14,15] monitoring technologies have been recognized by experts and scholars at home and abroad, and are widely used to monitor the early warning signals of ground pressure and other dynamic disasters. Based on the high-precision microseismic monitoring systems, Wang et al. continuously monitored the coalmines in China, comprehensively analyzed the microseismic signals, and established a rockburst prediction model, which provided the basis for early risk warning in the mining process [16]. Li et al. analyzed the characteristics of microseismic signal waves, effectively identified the precursor information of coal and rock dynamic disasters, and installed the microseismic and electromagnetic radiation monitoring systems in coalmines to monitor rockbursts. Through the comprehensive analysis of microseismic signal energy, microseismic signal event number, and electromagnetic radiation intensity, the prediction accuracy of rockbursts was improved [17,18]. Feng et al. proposed a dynamic early warning technology for the development process of tunnel rockbursts based on microseismic monitoring [19]. He et al. evaluated and predicted rockbursts through dynamic and static stress analysis based on MS monitoring [20]. Li established a theoretical model of acoustic emission propagation under the conditions of different robustness coefficients, and based on this, proposed the application conditions of acoustic emission for coal and rock dynamic disaster monitoring [21]. Wen used PFCD to analyze the acoustic emission characteristics of the damage evolution of different coal samples, and obtained four stages of the damage evolution of the coal and rock samples, so as to provide a basis for preventing coal and rock damage [22]. Gong embedded the acoustic emission signals obtained from rockburst experiments into time dynamics and analyzed the frequency shift phenomenon of rockburst precursor waves by using singular spectrum analysis (SSA) [23]. Based on the analysis and research of electromagnetic radiation monitoring data in Qianqiu Coal Mine, Song obtained a positive correlation between electromagnetic radiation and fracture of coal and rock, which provided a basis for early warning of rockbursts [24]. Mutke presented a new standard for risk assessment of rockbursts in longwall mining based on the seismic activity and features of hard coal seams in Poland [25]. Some Chinese scientific research teams headed by scholars such as Pan, Qi, Dou, Jiang, He, Wang, etc., and their scientific research on experiments, equipment development, and field practice have promoted the development of rockburst monitoring and early warning in China’s coalmines. Jiang et al. studied the classification and early warning methods of structurally controlled rockbursts by using the microseismic monitoring method and proposed two types of pressurization and decompression [26]. Xia et al. improved five risk prediction indicators based on microseismic monitoring technology, which can better predict the occurrence of rockbursts [27]. Pan et al. used a charge induction test system to analyze the charge induction laws of the coal and rock fracture process and carried out field tests. The results show that it can predict the occurrence of impact risk [28]. He and Wang started by revealing the phenomenon of electromagnetic radiation in coal and rock, studied the mechanism of electromagnetic radiation in loaded coal and rock, established an early warning model of coal and rock dynamic disasters, and had a wide range of applications in the prediction of rockbursts [29].

MS, EMR, and AE each have advantages in predicting impact pressure hazards. The test results greatly improve the understanding of the development process and monitoring and early warning of impact ground pressure. However, due to the different mechanisms of each physical monitoring method, there are inherent advantages and disadvantages. For example, the microseismic monitoring [30,31] range is large, which can monitor the location, intensity, and energy of rockbursts in real time, and can achieve more accurate positioning of rockbursts (the error is ±20 m in the horizontal direction and ±40 m in the vertical direction), but the prediction ability is weak. It is difficult to determine the risk of rockbursts by using the microseismic activity laws. Acoustic emission [32] monitoring technology has high positioning accuracy, but the monitoring range is small and it is easily affected by the noise of mine site operations, so the application effect is not ideal in practice. Different from microseismic monitoring and acoustic emission monitoring, electromagnetic radiation monitoring technology realizes real-time monitoring and early warning of rockbursts by non-contact monitoring of electromagnetic radiation precursor signals generated by surrounding rock ruptures caused by mining activities. Compared with microseismic and acoustic emission, the rockburst precursor of electromagnetic radiation is obvious, and it is sensitive to the stress distribution and change characteristics of the surrounding rock, but it is easily affected by the interference of underground electromagnetic fields (electromechanical and electrical equipment).

In order to improve the accuracy of early warning of rockburst hazards and minimize the errors caused by the inherent defects of monitoring methods, at present, nearly all mines with severe rockburst disasters in China have adopted multi-method and multi-indicator monitoring such as microseismic, acoustic emission, and electromagnetic radiation. The fact shows that the multi-indicator monitoring method has indeed improved the accuracy of early warning of rockburst hazards and played an active role in reducing the hazard degree of rockburst disasters.

However, there is still a certain gap compared with the real early warning and accurate forecast of rockburst disasters. In order to better monitor and early warning of rockburst disasters, this paper proposes an acoustic-electromagnetic synchronous testing technology for rockburst disasters, which organically integrates electromagnetic radiation technology and acoustic emission technology. At the same time, the YDD16 coal and rock dynamic disaster acoustic-electromagnetic monitoring instrument and data processing analysis software have been developed, and field test research on the acoustic-electromagnetic synchronous monitoring and early warning of rockburst has been carried out at the Fushun Laohutai Coal Mine excavation site, to improve the scientificity, timeliness, and reliability of impact pressure monitoring and early warning, in order to provide a certain reference for multi-indicator monitoring and early warning of rockbursts. The research results provide an important reference for predicting coalmine impact ground pressure risk under similar engineering geological conditions.

## 2. Prediction of Impact Hazard in Mining Site

### 2.1. Mine Overview

The Laohutai minefield is located in the southern part of Fushun City, Liaoning province, with geographical coordinates: 123°54′42″~123°58′17″ east longitude and 41°51′07″~41°52′10″ north latitude, as shown in Figure 1. The western part of the minefield is adjacent to the west open-pit minefield with a mining area coordinate of E3450m, and the eastern part is adjacent to the old Longfeng minefield with a mining area coordinate of E8400m. It starts from the coal seam outcrop in the south and ends at the F1 and F18 faults in the north. The east–west strike length is 4.8 km, the north–south width is about 2 km, and the minefield area is about 10 km^2^. There are two layers of coal in the minefield. The one-layer coal is the main mining coal seam. The three-layer (group B coal or lower coal) is mostly invaded by magma or transformed into shale. The coal seam is extremely unstable. The mine adopts the inclined shaft combined with the horizontal roadway development mode. All working faces are mined with comprehensive mechanized caving mining technology. The Laohutai mine is a typical rockburst mine; the main reason for the occurrence of rockburst is caused by self-weight stress, structural stress, and additional stress caused by mining after entering the critical mining depth; that is, it belongs to gravity, structure, and additional stress composite rockburst.

With the continuous increase of mining depths, rockburst disasters are becoming more and more serious.

### 2.2. Statistics of Rockburst Accidents

The two main mining working faces are 73005# and 83003#. In 2014, there were 58 high-energy mine earthquake events that caused obvious tremors on the working face, resulting in 8 rockburst accidents; among them, the 73005# working face occurred 12 times, accounting for 20.7% of the total; the 83003# working face occurred 46 times, accounting for 79.3%. Most of the rockburst accidents occurred near the fault, accounting for 93.1% of the total, and mainly concentrated in the high-stress areas beyond 30 m, accounting for 96.5%; the occurrence time was mainly related to the mining arrangement of the working face such as the advance degree of the mining face, and there was no significant difference distribution on the specific shift, which was relatively uniform.

An analysis of large energy impact events throughout 2014 combined with information on coalmine geological conditions, mining technology, and support methods. The main influencing factors on rockburst in the Laohutai Coal Mine include: large faults and folds, especially the presence of the F7 and F25 faults in the central and eastern regions, which cause tectonic stress concentration; the main coal seam having a tendency to impact and having high hardness (f = 1.5~3.0); the mining depth exceeding the critical depth of 580 m; and the goaf and coal pillars produced by comprehensive mechanized caving mining being able to easily lead to the superimposition of mining stress, which is reflected in the damage of the rockburst mainly in the bottom heave.

### 2.3. Monitoring Equipment and Layout

#### 2.3.1. Microseismic Monitoring System

The microseismic monitoring system is an integrated digital transmission signal system for seismic monitoring and location for microseismic risk assessment. The ARAMIS M/E microseismic monitoring system is mainly composed of a downhole seismometer, ground central station, and data recording and processing server. Microseismic monitoring is carried out on the mining area where the working face is located, and the vibration events with vibration energy greater than 100 J, a frequency range of 0–150 Hz, and lower than 100 dB are monitored. The energy and event counts are automatically calculated to determine the impact of the dangerous area and realize disaster warnings. 

The microseismic monitoring area of the Laohutai mine is divided into two parts: east and west. Both monitoring areas can change with the change of mine stope and move the seismometer at any time. The ARAMIS M/E monitoring system is 16 channels. The downhole vibration pickups are placed in each monitoring area according to their size, and the ARAMIS M/E system handles the data collecting, transmission, and analysis (the vibration pickup has not moved in the field) [33]. The structure of the microseismic monitoring system is shown in Figure 2.

#### 2.3.2. Acoustic Emission and Electromagnetic Radiation Monitoring System

The YDD16 coal and rock dynamic disaster acoustic-electromagnetic monitor is a portable, four-channel, multi-signal, non-contact directional monitoring instrument, composed of a host and a variety of signal (acoustic emission, ultrasonic, electromagnetic radiation, ultra-low frequency) sensors that can be connected (as shown in Figure 3), which can carry out mobile, multi-signal synchronization, and large-scale regional monitoring of coal and rock dynamic disasters such as rockbursts. At the same time, it is equipped with special data processing and analysis software (as shown in Figure 4), which has the functions of communication, data and chart display, original waveform acquisition and display, pulse statistical analysis, energy statistical analysis, event statistical analysis, and early warning and reporting. The monitoring data are processed to determine the regional and dynamic laws of danger and give early warning.

The electromagnetic radiation sensor is a high-sensitivity, wide-band (1~500 kHz), non-contact test. During the test, the electromagnetic radiation monitoring antenna is suspended toward the center of the monitored coal area, and the opening seam is facing the interior of the coal, avoiding the interference source as far as possible. The acoustic emission sensor is a contact test, and the monitoring frequency is 1 kHz~5.5 kHz. It needs to be fixed on the roadway bolt during monitoring, and the bolt is used as the waveguide rod, to ensure that the sensor and the bolt make good contact (as shown in Figure 5).

#### 2.3.3. Layout of Acoustic-Electromagnetic Monitoring Points

The 73005# working face is tested once a day in the morning shift. At present, it is the finishing work, which will be finished soon. The test location is the pedestrian conveying channel, and the test spacing is 15 m.

The monitoring position of the 83003# working face is the pedestrian conveying channel and the return air trough, and the test interval is 30 m. The test is performed once a day, and the test is performed in the morning shift. The test is conducted once a day during the morning shift.

### 2.4. Monitoring Results and Precursor Laws

Through the analysis of many typical rockburst events that occurred in the working face of 73005# and 83003# in 2014, it is found that the microseismic, acoustic emission and electromagnetic radiation signals before the rockburst in the Laohutai Coal Mine have obvious abnormal characteristics and precursor response laws. Specific events are described below.

#### 2.4.1. Microseismic Monitoring Results

(1)24 February 2014 rockburst event 73005# working face

It can be seen from Figure 6a that the changing trend of microseismic energy and frequency remained synchronized. After the extremely low value appeared on 17 February, the microseismic energy value and frequency increased sharply on 18 February, and the microseismic energy value and frequency remained at a high level in the next few days. The microseismic signal showed a local high value on 18 February, then decreased slightly on 19 February, increased again on 20 February, and then decreased continuously. The impact occurred on 24 February, resulting in damage to the four-inch pipe valve at the 73005# water injection pump and a small amount of plasma. This event can be considered an “inverted V-shape”.

(2)12 July 2014 rockburst event 83003# working face

As can be seen from Figure 6b, from 1 to 8 July, the daily total energy remained at a high level of the 4th power, and the energy in the coal rock mass was continuously released, but the daily frequency showed a downward trend, indicating that the energy value of a single event was rising. On the 9th, a low value suddenly appeared, followed by a vibration-type mine earthquake on the 10th, and the energy value increased rapidly. After a short release of energy, the impact occurred again on the 12th. After the 10th, the daily total energy was generally stable, but the daily frequency was gradually decreasing, that is, the energy value of a single event was rising. Therefore, this impact event can be classified as a “V-type” event. The impact of the 12th day caused three roofs and two sides of the anchor spray coating to crack at 70–90 m of the 73005# pedestrian transport channel, and the iron chain between 48# and 49# of the hydraulic support of the 83003# return air roadway was broken.

Before the occurrence of a rockburst, the lowest point of microseismic energy appears at the lowest point of frequency change, because during the increase of daily total energy and frequency, the number of microseismic events increases, energy is released, and the internal energy of coal and rock mass is not enough to destroy the stress balance of coal and rock mass, resulting in the reduction of daily total energy and frequency. With the accumulation of energy, under the action of external disturbance rockburst is induced and energy is released, increasing total daily energy.

#### 2.4.2. Acoustic-Electromagnetic Monitoring Results

(1)24 February, 2014 rockburst event-73005# working face

It can be seen from Figure 7 that the acoustic emission ringing and electromagnetic radiation intensity change trends monitored on the site of 73005# working face are in good agreement. They were on an upward trend from the 19th to the 23rd, and a sudden increase occurred on the 23rd. According to the analysis, the micro-crack friction intensifies and causes small damage, which leads to the increase of acoustic emission ringing number and electromagnetic radiation intensity. Acoustic emission amplitude and electromagnetic radiation pulse show an overall “inverted V-shaped” trend. The high-value state in the early stage showed an upward trend and then began to decline after the 23rd. The amplitude of acoustic emission declined suddenly from 1200 mV to 250 mV, and the electromagnetic radiation pulse dropped from 160 kHz to 45 kHz, which was a significant drop. Then, a rockburst occurred on the 24th. 

(2)12 July 2014 rockburst event 83003# working face

The acoustic emission signal change curve obtained through the on-site monitoring of the 83003# working face is shown in Figure 8a. The acoustic emission signal showed a V-shaped change trend as a whole. The acoustic emission ringing showed a downward trend from 3 to 6 July, reached the minimum value on the 6th, then increased for five consecutive days and reached the peak on the 11th; the amplitude of acoustic emission responded one day ahead of the ringing, dropped to the low value on the 5th, and then continued to rise, reaching the maximum value on the 10th, and finally, the rockburst occurred on the 12th. The fluctuating trend of electromagnetic radiation is shown in Figure 8b. The electromagnetic radiation intensity is consistent with the pulse change trend. It continued to rise from 5 July and reached a peak on the 9th. The increase rate exceeded 200%. Then, the coal and rock structure was destroyed and energy was released. The intensity and pulse of electromagnetic radiation dropped sharply, and the trend of change showed an inverted V-shape. During the decline, a certain amount of energy was accumulated and a rockburst occurred on the 12th.

#### 2.4.3. Precursor Laws

(1)The changing trend of microseismic energy and frequency in Laohutai Coal Mine has a good consistency, which can reflect the failure strength and frequency of coal rock. The frequency generally increases with the increase of total energy and decreases with the decrease of total energy, showing a positive correlation. The microseismic signal has obvious fluctuation before the rockburst event, and there is a very low value within five days before the impact. The common feature is that there is an upward stage, and the rockburst occurs during the upward process. Therefore, if the microseismic signal value shows a continuous upward or upward trend in the process of change, and there is a very low value in the process of change, it indicates that the possibility of impact events is greater, to which special attention should be paid.(2)Summarizing the acoustic-electromagnetic response laws of rockburst events, it can be seen that the overall change trend of electromagnetic radiation and acoustic emission signals in the Laohutai Coal Mine before rockburst events can be divided into two basic forms: rising type and inverted V-shaped. The rising type shows the rising trend of the acoustic-electromagnetic signal exceeding four consecutive days, while the inverted V-shaped shows that the acoustic-electromagnetic signal rises to the peak state and the rockburst occurs in the descending process, which is closely related to the heterogeneous structure of coal. In contrast, the acoustic emission signal changes more diverse, and the electromagnetic radiation signal is mainly represented by an inverted V-shaped.

## 3. The Method and Application of Acoustic-Electromagnetic Early Warning for Rockbursts

### 3.1. Theoretical Basis

In 1941, seismological experts Gutenberg and Richter proposed the famous earthquake magnitude–frequency relationship, which can be referred to as the G–R relationship, by studying the characteristics of California earthquake activity [34]: (1)$$\mathrm{lg}N=a_{0}-b_{0}M$$
where M is magnitude; *N* is the total number of earthquakes with a magnitude greater than M; *a*_0_ and *b*_0_ are constant.

The research shows that the relationship between the frequency and energy of earthquakes (mine earthquakes, rockbursts) induced by human mining activities and natural earthquakes follows the G–R relationship. The G–R relationship is universal. The G–R relationship and *b* value can be used as important indicators for studying mine earthquakes, rockbursts, and other activities.

In this paper, the fluctuation theory is applied to the monitoring and early warning of rockbursts, and the dynamic variation of the acoustic-electromagnetic signal in unit time is introduced, which is the index of the acoustic-electromagnetic time fluctuation gradient. The theoretical basis is that electromagnetic radiation and acoustic emission signals are the release of electromagnetic waves and elastic waves during the deformation and destruction process of coal and rock masses under the influence of mining, and have a corresponding relationship with coal and rock mass loads. When the external stress state changes greatly, the acoustic-electromagnetic signal will fluctuate obviously. Once the fluctuation degree exceeds a certain range, it means that the possibility of danger increases. Obviously, the non-equilibrium change of energy in coal and rock mass fluctuates continuously with time and has significant time characteristics. Therefore, the time fluctuation gradient of the acoustic-electromagnetic signal can be applied to the monitoring and early warning of rockbursts. It is specifically expressed as follows:(2)$$\Delta Y_{t}=\left(Y_{t}-Y_{t-n}\right)/n$$
where ∆*Y_t_* is the time fluctuation gradient of parameter *Y* at time *t*, *Y* is the electromagnetic intensity, electromagnetic pulse, acoustic emission ringing, or acoustic emission amplitude, *Y_t_* and *Y_t_*_−*n*_ are the value of parameter *Y* at time *t* and *t* − *n*, respectively, and *n* is the time interval.

### 3.2. Hazard Discrimination Index

The time fluctuation gradient is introduced as the characteristic index of the acoustic-electromagnetic signal. According to the monitoring data of the YDD16 acoustic electric monitor and combined with wave theory, electromagnetic intensity fluctuation gradient ∆E, electromagnetic pulse fluctuation gradient ∆N, acoustic emission ringing fluctuation gradient ∆R, and acoustic emission amplitude fluctuation gradient ∆A are selected as early warning and discrimination indexes of electromagnetic radiation and acoustic emission signal, respectively. At the same time, the variation rule of the microseismic *b* value index of the ARAMIS M/E microseismic monitoring system is also referred to. The microseismic data and the acoustic-electromagnetic data of the two working faces during a certain monitoring period are processed.

According to the calculation principle of the least square method, the calculation formula of the *b* value is obtained as follows:(3)$$b=\left\{\begin{array}{l} \frac{\sum_{i=1}^{m}\mathrm{lg}E_{i}\sum_{i=1}^{m}\mathrm{lg}N_{i}-m\sum_{i=1}^{m}\mathrm{lg}E_{i}\mathrm{lg}N_{i}}{m\sum_{i=1}^{m}{\mathrm{lg}}^{2}E_{i}^{}-{\left(\sum_{i=1}^{m}\mathrm{lg}E_{i}\right)}^{2}}\\ 0 \end{array}\right.$$
where *m* is the total number of energy classifications.

Considering that the number of microseismic events less than the initial microseismic energy may be incompletely recorded for various reasons, and the number of high-energy microseisms significantly impacts the *b*-value, the microseismic events within a certain energy range are usually selected to estimate the *b*-value. In addition, in order to reduce the error as much as possible, the microseismic energy is classified to ensure that the data in the energy classification are close to reality.

In this paper, the energy lower limit *E*_0_ of microseismic data is 1 × 104 J, ∆M_0_ = lg*E* = 0.2, time window *T* is 15 d, and Sliding step ∆*T* is 1 d. The specific calculation results are shown in Figure 9.

The Figure 9 shows the fluctuation curve of b characteristic value with time in the 73005# working face and the 83003# working face according to microseismic monitoring data. According to the Figure 9, it is found that most of the rockbursts and high-energy mine earthquake events occur during the period of low *b* value. If the microseismic *b* value drops to the lower limit of each reference or is within the range of fluctuating peaks and valleys, the risk of coal-rock dynamic disasters is more significant, to which special attention should be paid.

According to the monitoring data of the YDD16 acoustic electric monitor, electromagnetic intensity fluctuation gradient ∆E, electromagnetic pulse fluctuation gradient ∆N, acoustic emission ringing fluctuation gradient ∆R, and acoustic emission amplitude fluctuation gradient ∆A are selected as early warning and discrimination indexes of electromagnetic radiation and acoustic emission signal, respectively. Take the 73005# working face as an example; the acoustic-electromagnetic time fluctuation gradient curve is shown in Figure 10.

By analyzing the Figure 10, it is found that ∆E, ∆N, ∆R, and ∆A fluctuate up and down in a certain range around the value of 0. The temporal variation rules of four indexes, namely, the electromagnetic intensity fluctuation gradient ∆E, the electromagnetic pulse fluctuation gradient ∆N, the acoustic emission ringing fluctuation gradient ∆R, and the acoustic emission amplitude fluctuation gradient ∆A, are analyzed, and their corresponding reference critical values are determined. Specifically: ∆E_0_ = ±280 mV/d, ∆N_0_ = ±200 kHz, ∆R_0_ = ±140 × 103 pieces/d, ∆A_10_ = ±280 mV/d. The fluctuation of the over-range indicates that the cracks in the coal and rock mass develop and penetrate, and obvious structural damage occurs, suggesting that the possibility of dynamic disasters in the short term is greater, and attention should be paid to prevention.

### 3.3. Hazard Warning Guidelines

Multi-source information fusion can properly fuse the redundant, complementary, and even conflicting information of the same or different types of multiple sensors in the system with a certain rule, so as to achieve a consistent description of the real situation of the observation area and reflect the essence of things more truly and comprehensively, so as to make up for the shortcomings of a single monitoring method or a single sensor. D-S evidence theory is an information fusion method with simple application and strong independence [35].

D-S theory is to use the combination rule to fuse several incomplete conflicting belief functions based on different evidence in the same recognition framework to calculate a belief function. Evidence theory has its unique advantages: (1) Evidence theory has a strong theoretical basis, which can deal with both the uncertainty caused by randomness and the uncertainty caused by fuzziness; (2) Evidence theory can distinguish between unknown and uncertain; (3) Evidence theory does not need prior probability and conditional probability density, so it is convenient for application. The specific definition is as follows: Given that ∀A⊆Θ and A≠∅, *m*_1_ and *m*_2_ are two mass functions on Θ, respectively, and the Dempster synthesis rule is: [36,37]
(4)$$m_{1}⊕m_{2}\left(A\right)=\frac{1}{K}\sum_{B\cap C=A}m_{1}\left(B\right)\cdot m_{2}\left(C\right)$$
where *K* is the normalization constant:(5)$$K=\sum_{B\cap C\neq \Phi }m_{1}\left(B\right)\cdot m_{2}\left(C\right)=1-\sum_{B\cap C=\Phi }m_{1}\left(B\right)\cdot m_{2}\left(C\right)$$
the combination rules of multiple trust functions are as follows:

Suppose there are a finite number of mass functions on the recognition framework Θ, which are m_1_, *m*_2_, …, *m_n_*, ∀A⊆Θ and A≠∅, and the synthesis rule is:(6)$$\left(m_{1}⊕m_{2}⊕\ldots ⊕m_{n}\right)\left(A\right)=\frac{1}{K}\sum_{A_{1}\cap A_{2}\cap \ldots \cap A_{n}=A}m_{1}\left(A_{1}\right)\cdot m_{2}\left(A_{2}\right)\ldots m_{n}\left(A_{n}\right)$$
(7)$$K=\sum_{A_{1}\cap A_{2}\cap \ldots \cap A_{n}\neq \Phi }m_{1}\left(A_{1}\right)\cdot m_{2}\left(A_{2}\right)\ldots m_{n}\left(A_{n}\right)=1-\sum_{A_{1}\cap A_{2}\cap \ldots \cap A_{n}=\Phi }m_{1}\left(A_{1}\right)\cdot m_{2}\left(A_{2}\right)\ldots m_{n}\left(A_{n}\right)$$

According to the actual situation of the Laohutai Coal Mine, the identification framework of the rockburst algorithm model system is Θ = {Θ1, Θ2, Θ3, Θ4}, [38] that is, the specific identification framework is Θ = {Level Ⅰ, Level Ⅱ, Level Ⅲ, Level Ⅳ}. The comprehensive discriminant criteria are as follows (Table 1):

Different levels of risk require different attention and response measures, and the significance and specific description of hazard determination results are shown in the following table (Table 2) [39]:

In order to improve the reliability and accuracy of hazard early warning, according to the D-S evidence theory, comprehensive consideration is made on the basis of the respective hazard discrimination results of the five indicators. Therefore, the comprehensive index D_i_ of acoustic-electromagnetic early warning, evidence body of identification framework is put forward, *i* = 1, 2, …, *n*. In $D_{i}=\left\{D_{1},D_{2},D_{3},D_{4},D_{5}\right\}$, *D_i_* stands for *b* value, ∆E, ∆N, ∆R, and ∆A, respectively.

According to the variation trend of the microseismic *b* value with time, it is found that the *b* value is lower than the reference threshold, and the smaller the value, the greater the risk. Therefore, the following membership functions are used to calculate the probability value of microseismic *b* value risk level:(8)$$\mu_{i}\left(b_{j}\right)=\frac{1}{2}\left(1-\frac{b_{ij}-b_{i0}}{Max\left(\left|b_{ij}-b_{i0}\right|\right)}\right)$$

According to the study of the acoustic and electrical signals of the two working faces, it is found that the overall acoustic-electromagnetic time fluctuation gradient curve fluctuates around the value of 0. The farther away from the value of 0, the greater the possibility of danger. Therefore, the probability value calculation of the hazard level of the acoustic-electromagnetic time fluctuation gradient adopts the following membership function:(9)$$\mu_{i}\left(\Delta Y_{j}\right)=\frac{1}{2}\left(1+\frac{\left|\Delta Y_{ij}\right|-\left|\Delta Y_{i0}\right|}{Max\left(\left|\Delta Y_{ij}-\left|\Delta Y_{i0}\right|\right|\right)}\right)$$

Among them, *I* = 1, 2, representing the 73005# working face and the 83003# working face, respectively; *j* = 1, 2, 3, …, representing the date corresponding to the data; b*_i_*_0_ and ∆*Y_i_*_0_, respectively, represent the *b* value of each working face and the reference critical value of time fluctuation gradient.

The reference critical values of each working face are as follows:

*b*_10_ = 0.40, ∆E_10_ = ±280 mV/d, ∆N_10_ = ±200 KHz, ∆R_10_ = ±140 × 103 pieces/d, ∆A_10_ = ±280 mV/d; *b*_20_ = 0.33, ∆E_20_ = ±190 mV/d, ∆N_20_ = ±160 KHz, ∆R_20_ = ±30 × 103 pieces/d, ∆A_20_ = ±200 mV/d.

After the probabilities of hazard levels predicted by each evidence body are known, the basic probabilities of the main propositions need to be assigned within the recognition framework. In order to avoid conflicts of evidence, ensure that the basic probability values of the main propositions are between 0.30 and 0.40. The specific assignment rules are as follows:(10)$$m_{i}\left(x_{j}\right)=\left\{\begin{array}{l} \left(\mu_{i}\left(x_{j}\right)-0\right)\cdot 0.4+0.3\\ \left(\mu_{i}\left(x_{j}\right)-0.25\right)\cdot 0.4+0.3\\ \left(\mu_{i}\left(x_{j}\right)-0.5\right)\cdot 0.4+0.3\\ \left(\mu_{i}\left(x_{j}\right)-0.75\right)\cdot 0.4+0.3\\ 0.35 \end{array}\right.\begin{array}{l} \left(0\leq \mu_{i}\left(x_{j}\right)<0.25\right)\\ \left(0.25<\mu_{i}\left(x_{j}\right)<0.5\right)\\ \left(0.5<\mu_{i}\left(x_{j}\right)<0.75\right)\\ \left(0.75<\mu_{i}\left(x_{j}\right)\leq 1\right)\\ \left(\mu_{i}\left(x_{j}\right)=0.25,0.5,0.75\right) \end{array}$$
where *j* is time, *m_i_*(*x_j_*) represents the basic probability assignment of the main proposition corresponding to *x* evidence body in the i working face, and *x* represents *b* value, ∆E, ∆N, ∆R, and ∆A, respectively.

When the basic probability value is at the dividing point, namely, 0.25, 0.5, 0.75, it shows that the division of danger degree is uncertain and fuzzy. In order to approach reality as much as possible, the basic probability value of adjacent propositions is set as 0.35.

After the basic probability assignment of each evidence body index is completed, the data fusion is carried out according to the evidence combination rules shown in Equations (4) and (6). The final multi-index data fusion result is judged by referring to Table 1. When it reaches level II or shows an obvious growth trend for three consecutive days, it can give an early warning.

### 3.4. Early Warning Practice

After establishing the early warning method of the rockburst hazard based on the fluctuation gradient, the 63007# working face is selected as the research object to conduct practical research on the rockburst hazard acoustic and electrical early warning.

The 63007# working face is located at −630 level with a ground elevation of 67.3 m~84.7 m and an underground elevation of −587 m~−641 m. The working face is adjacent to the 55003# fully mechanized caving face in the east, 63005# fully mechanized caving face in the west, 58003#-1 planned face in the south, minefield boundary in the north, 58003#, 55001#, and 63002# fully mechanized caving faces in the upper part, and five layered coal seam and coal seam floors in the lower part, which has little impact on the driving roadway.

At 0:47 on 8 July 2014, an impact occurred in the roadway 48 m north of the 63007# transportation tunnel. The source location (36,625, 79,429, −537) occurred in the coal seam. The microseismic energy was 4.5 × 10^6^ J. The microseismic magnitude was level 3.1, causing 20 hydraulic mono columns to be skewed at 30–40 m along the transportation trough of 63007#, and the bottom of the bottom plate was 200 mm. Figure 11 shows the changing trend of the five indicators before the occurrence of the rockburst and the final fusion results.

It can be seen that the overall change trend of the five indicators was quite different, the overall value of b was relatively high, and it continued to decrease from the 1st to the 7th, and the risk gradually increased. The lowest value on the 7th also reached 0.48, which was close to the danger warning line of 0.5. The electromagnetic intensity fluctuation gradient ∆E was in the process of fluctuation and remained above 0.38. It reached the level II early warning on the 2nd and 4th, rose for three consecutive days from the 5th, and reached 0.74 on the 7th, which was close to the most dangerous level I early warning. The electromagnetic pulse fluctuation gradient ∆N was relatively stable before the 5th and remained around 0.3. It suddenly increased to 0.73 on the 6th and continued to increase to 0.91 on the next day, exceeding the level I warning level of 0.75. The fluctuation gradient ∆R of acoustic emission ringing was the most violent, which was only 0.03 on the 1st, increased to 0.37 on the 2nd, then dropped sharply to 0.12 on the next two days, and then continued to rise, reaching a peak value of 0.82 on the 7th. The whole change process crossed the warning level I from level IV.

The fluctuation gradient ∆A of acoustic emission amplitude was relatively stable, fluctuating slightly around 0.48 during the period from 1st to 6th, increasing to 0.72 on the 7th, and reaching the more dangerous warning level of level II for three consecutive days on the 5th to 7th.

According to the resulting curve of data fusion, the comprehensive prediction value fluctuated in the range of 0.28~0.37 from the 1st to the 5th, which belonged to the safety warning level of level III. It rose for four consecutive days from the 4th and increased to 0.56 on the 6th. According to the early warning criteria, the early warning started and reached the peak value of 0.7 on the 7th. The early warning continued, then the rockburst occurred on the 8th, causing the hydraulic support deflection and floor heave.

After the occurrence of the rockburst, the comprehensive early warning value decreased, and the electromagnetic pulse fluctuation gradient ∆N, the acoustic emission ringing fluctuation gradient ∆R, and the acoustic emission amplitude fluctuation gradient ∆A all decreased significantly in the same period; and the *b* value increased, indicating that the risk decreased; but there are still three index values more than 0.5; and the comprehensive early warning value is within the scope of level II early warning, so-anti scour measures should continue to be taken to ensure safe production.

To sum up, multi-information fusion early warning results of rockbursts based on D-S evidence theory are completely consistent with field rockburst instability events; the four indicators of ∆E, ∆N, ∆R, and ∆A can achieve a synchronous response 1 to 5 days ahead of rockburst events. Compared with the electromagnetic radiation or acoustic emission single method, its early warning accuracy and reliability will be better, and it is suitable for advanced early warning of rockburst disasters.

## 4. Joint Early Warning Method

Based on the three monitoring methods of MS, EMR, and AE, a multi-index monitoring and early warning method for accurately predicting rockburst is established. The method consists of the following five steps (Figure 12). 

(1)Installation of monitoring equipment, design, and data collection. According to the geological conditions of Fushun Laohutai Coal Mine, monitoring equipment is installed and a monitoring scheme is designed before mining. Ensure that all monitoring data of MS, EMR, and AE can be completely recorded during the mining process.(2)Rockburst precursor information. According to the monitoring and test data of microseismic, electromagnetic radiation, and acoustic emission collected and measured, the response characteristics of each rockburst disaster are analyzed.(3)Select and determine the early warning index of rockburst. Determine the appropriate microseismic and acoustic indicators, analyze their time series changes, and summarize their respective thresholds.(4)Identify the risk of rockburst. Based on the D-S evidence theory, the multi-parameter monitoring indicators are combined to determine the identification framework, obtain the evidence body, carry out decision fusion and decision analysis, and establish a multi-parameter monitoring and early warning model for rockbursts.(5)Application and verification. Combined with engineering practice, the effectiveness of the multi-index fusion early warning technology is verified.

## 5. Conclusions

The YDD16 acoustic-electromagnetic monitor is used to test the acoustic emission (AE) and electromagnetic radiation (EMR) signals before the occurrence of rockbursts in the working faces in the LaohutaiCoal Mine. Based on the fluctuation theory, the comprehensive early warning technology of multi-index for rockburst prediction in the Laohutai Coal Mine was put forward and applied into practice. 

The precursory laws of microseismic and acoustic signals of rockburst in the Laohutai Coal Mine have been analyzed. It is found that the correlation between energy and frequency parameters of microseismic signals in the Laohutai Coal Mine is good. The overall change trend of electromagnetic radiation and acoustic emission signals before rockburst events is divided into two basic forms: rising type and inverted V-type, and the signals have obvious fluctuation before rockburst. It is feasible to use acoustic emission-electromagnetic radiation-microseismic monitoring technology for comprehensive early warning of coalmine rockbursts.According to the seismic G–R relationship and the wave theory, the microseismic *b* value and the acoustic–electric time fluctuation gradient are determined as the main research indicators of rockburst monitoring and early warning, and the respective thresholds are obtained as the critical reference value for judging the risk degree of rockbursts. According to the D-S evidence theory, the above five indicators are used as evidence bodies for decision fusion and decision analysis, and a comprehensive early warning technology for multi-index information fusion of rockburst in Laohutai Coal Mine is established.The method is verified in the 63007# working face. The field application shows that the method can accurately identify and warn of the potential risk of rockburst in the working face and ensure the safe and efficient production of the working face, which has important guiding significance for the prevention and control of rockburst in coalmines in China.

## Figures and Tables

**Figure 1 ijerph-20-00392-f001:**
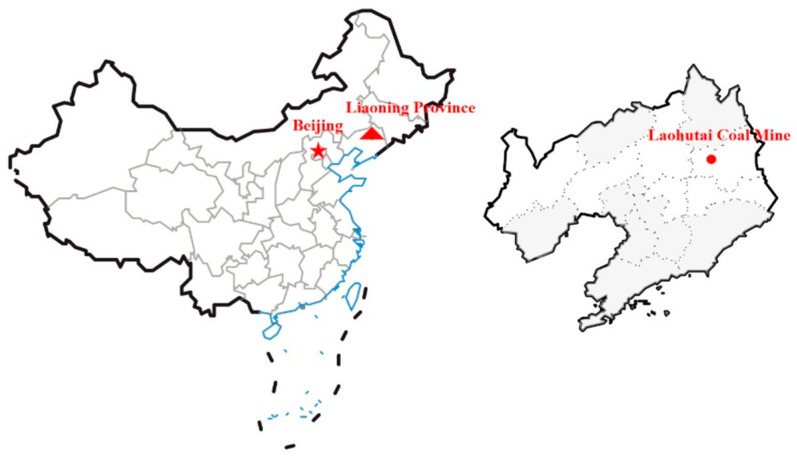
Geographical location of Laohutai Coal Mine.

**Figure 2 ijerph-20-00392-f002:**
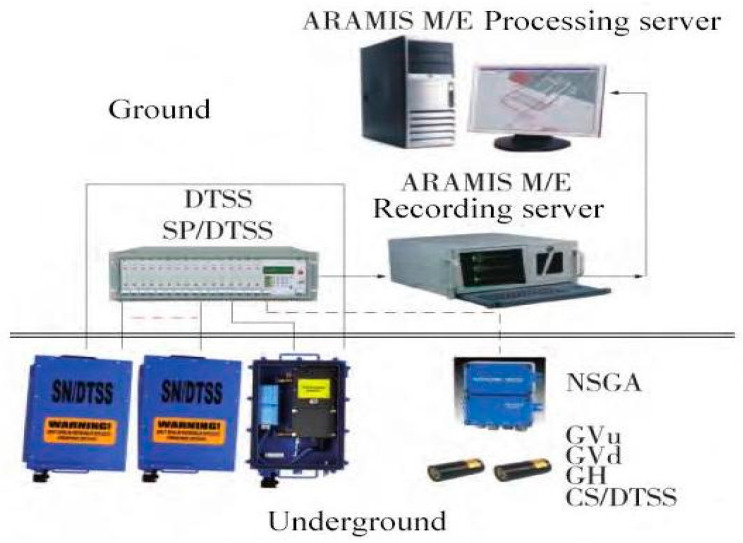
Microseismic system diagram.

**Figure 3 ijerph-20-00392-f003:**
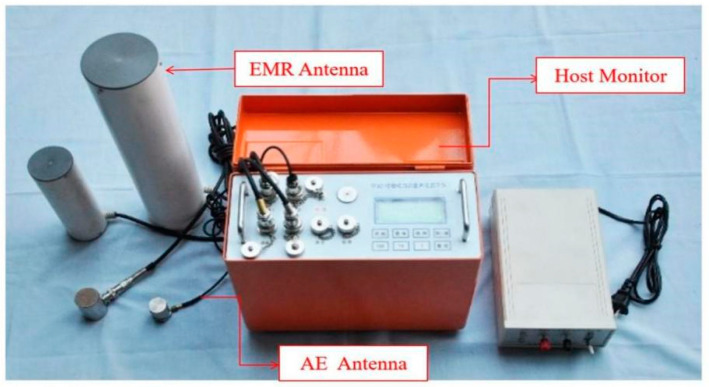
YDD16 acoustic-electromagnetic monitor.

**Figure 4 ijerph-20-00392-f004:**
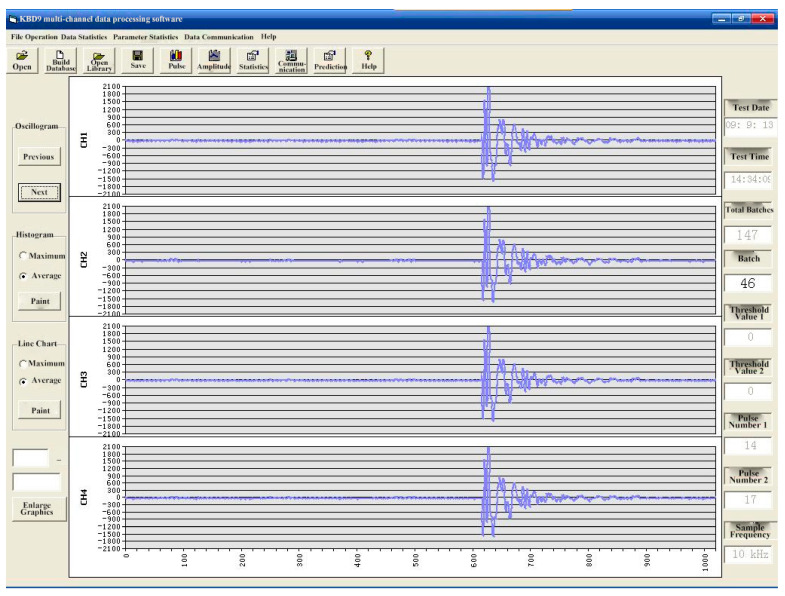
Main interface of data processing software in Chinese.

**Figure 5 ijerph-20-00392-f005:**
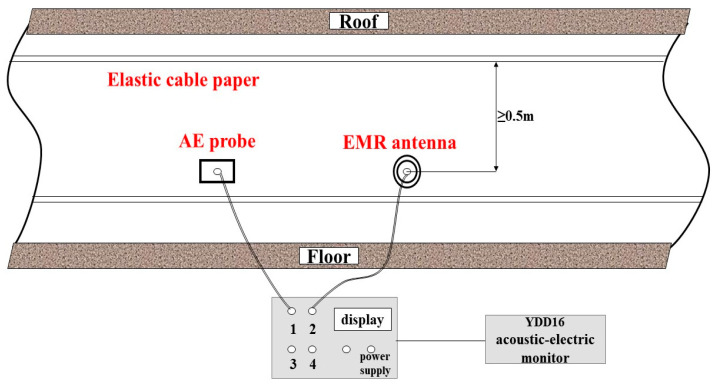
Use diagram of the YDD16 monitor.

**Figure 6 ijerph-20-00392-f006:**
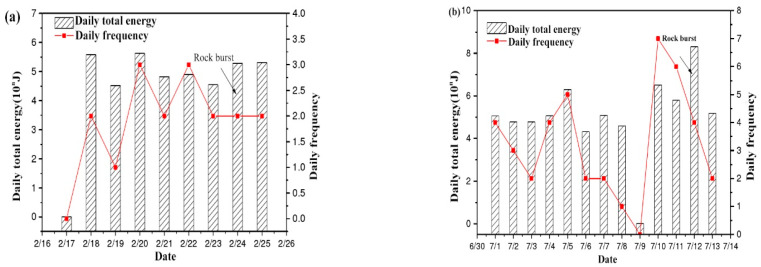
ARAMIS M/E microseismic rule of rockburst. (**a**) On 24 February 2014; (**b**) On 12 July 2014.

**Figure 7 ijerph-20-00392-f007:**
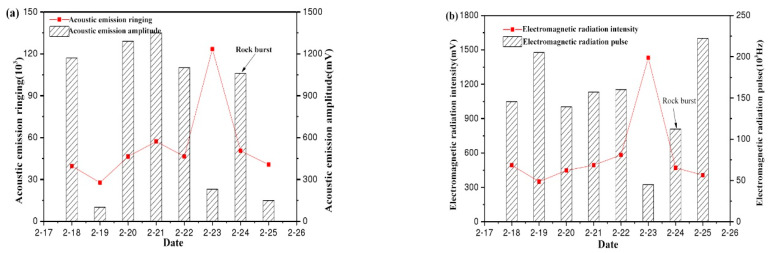
EMR and AE rule of rockburst disaster on 24 February 2014: (**a**) AE (**b**) EMR.

**Figure 8 ijerph-20-00392-f008:**
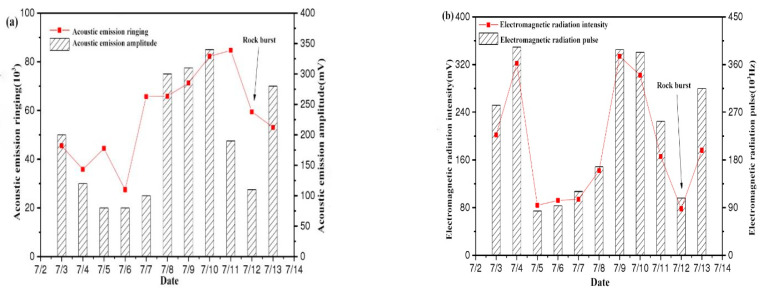
EMR and AE rule of rockburst disaster on 12 July 2014: (**a**) AE (**b**) EMR.

**Figure 9 ijerph-20-00392-f009:**
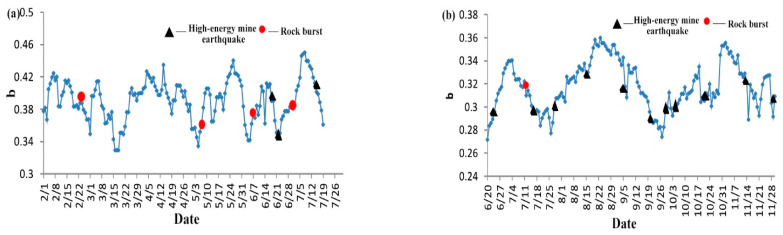
Microseismic *b* value change curve (**a**) 73005# (**b**) 83003#.

**Figure 10 ijerph-20-00392-f010:**
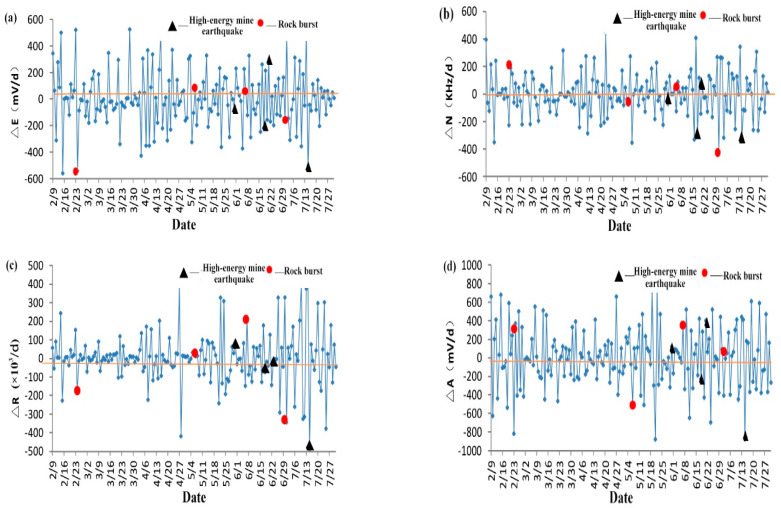
Time fluctuation gradient curve of 73005# working face. (**a**) ∆E. (**b**) ∆N. (**c**) ∆R. (**d**) ∆A.

**Figure 11 ijerph-20-00392-f011:**
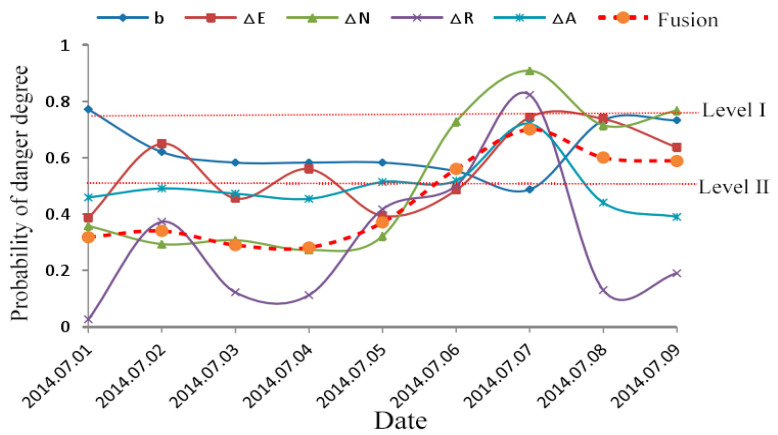
Prediction of rockburst on 8 July 2014.

**Figure 12 ijerph-20-00392-f012:**
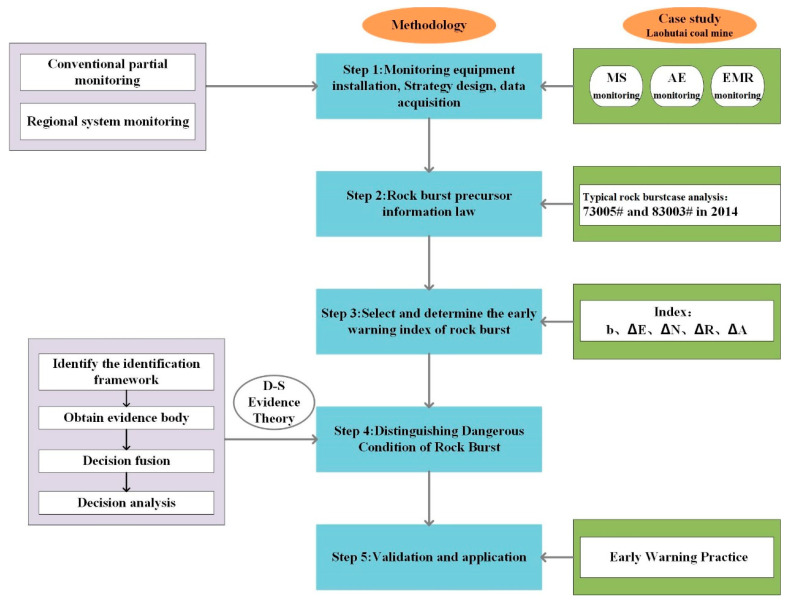
Flowchart of combined early warning method of rockburst.

**Table 1 ijerph-20-00392-t001:** Judging table of hazard level.

Hazard Level	Level Ⅰ	Level II	Level III	Level IV
Judgement standard	[0.75,1]	[0.5,0.75)	[0.25,0.5)	[0,0.25)

**Table 2 ijerph-20-00392-t002:** Meaning and statement of hazard level.

Hazard Level	Representative Meaning	Specific Instructions
Level Ⅰ	Dangerous	Regional and local prevention and control measures are taken and checked, and monitoring and forecasting are carried out at the same time. Excavation can only be carried out when the safety level is reached.
Level II	More dangerous	The prevention and control measures should be strengthened during excavation, and monitoring and forecasting are carried out at the same time. Excavation can only be carried out when the safety level is reached.
Level III	Safer	Strengthening the monitoring and forecasting of coal and rock dynamic disasters during mining
Level IV	Safe	Excavation work can proceed normally.

## Data Availability

Not applicable.

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
