# Peer review of "Multi-Index Geophysical Monitoring and Early Warning for Rockburst in Coalmine: A Case Study"

_ijerph, 2022, doi:10.3390/ijerph20010392_

Round 1
Reviewer 1 Report
The multi-index geophysical monitoring and early-warning method and application for coal burst in coalmine were conducted in this work. The content of the manuscript is well organized and the innovation is relatively strong. The manuscript is recommended to be accepted after minor revision. Some issues which should be addressed are listed below,
1) There are still some writing or grammatical mistakes in the manuscript, e.g., line 78, line 192.
2) The frequency ranges of acoustic emission, microseism and electromagnetic radiation monitoring are compatible with the signals, but the frequency ranges of the three types of signals are different. How should we consider the impact of these differences on the joint analyses?
3) Necessary references should be added to the derivation of some formulas.
4) The basis for determining the judging range of different disaster levels shall be supplemented.
5) The manuscript stated that compared with electromagnetic radiation or acoustic emission single method, its early warning accuracy and reliability will be better. Can the advantages of this joint analysis give quantitative analysis results? How much can the prediction success rate increase?
Author Response
Thank you for your letter and comments concerning our manuscript entitled “Multi-index Geophysical Monitoring and Early-warning for Coal Burst in Coalmine: A Case Study” (Manuscript Number: IJERPH-2083398). Those comments are all valuable and very helpful for revising and improving our paper, as well as the important guiding significance to our research. We have studied the comments carefully and made corrections, which we hope meet with approval. Revised portions are marked in red on the paper. The leading corrections in the paper and the responses to the reviewer’s comments are as following:
Responds to reviewer #1’s comments: The multi-index geophysical monitoring and early-warning method and application for coal burst in coalmine were conducted in this work. The content of the manuscript is well organized and the innovation is relatively strong. The manuscript is recommended to be accepted after minor revision. Some issues which should be addressed are listed below,
1) There are still some writing or grammatical mistakes in the manuscript, e.g., line 78, line 192.
Response 1: Thank you very much for your important comment. Mistakes in lines 78 and 192 have been corrected in the manuscript.
2) The frequency ranges of acoustic emission, microseism and electromagnetic radiation monitoring are compatible with the signals, but the frequency ranges of the three types of signals are different. How should we consider the impact of these differences on the joint analyses?
Response 2: Thank you very much for your important comment. In the practice of monitoring and early warning of rock burst in the coal mine, all kinds of real-time monitoring systems have different application scopes. The microseismic system mainly monitors and warns the rock burst with large energy in the whole mine, and supplements the monitoring system of electromagnetic radiation and acoustic emission in small areas with small energy. The activity law of mine dynamic disaster is predicted by monitoring the whole process of energy accumulation and release in coal and rock mass. Therefore, in this paper, DS evidence theory is introduced to multi-source information fusion for microseismic, electromagnetic radiation and acoustic emission systems in online operations. In this process, data processing, data mining, information fusion and monitoring and early warning are carried out. Integrate the early warning results of multiple monitoring systems to achieve distributed multi-point, regional automatic monitoring and hierarchical early warning of rock burst. Although the frequency ranges of the three signals are different, fusion processing can effectively reduce the impact of these differences on joint analysis.
3) Necessary references should be added to the derivation of some formulas.
Response 3: Thank you very much for your important comment. The references have been revised. Please see the manuscript.
4) The basis for determining the judging range of different disaster levels shall be supplemented.
Response 4: Thank you very much for your important comment. The determination of the judgment range of different disaster levels is based on the references, which have been identified above Table 2 of the manuscript. Please see the attachment.
5)The manuscript stated that compared with electromagnetic radiation or acoustic emission single method, its early warning accuracy and reliability will be better. Can the advantages of this joint analysis give quantitative analysis results? How much can the prediction success rate increase?
Response 5: Thank you very much for your important comment. Based on the multi-index information fusion early-warning technology established in the article, it was verified and applied in the 63007 # working face of the Laohutai Coal Mine. Due to the space limitation of the article, only the typical time of the rock burst on July 8, 2014 was selected for analysis and explanation. According to the actual data statistics, early warning has been given for the rock burst that occurred at 63007 # working face in 2014, and the large energy mine earthquake has also responded. There is no missing report, and the accuracy of early warning is 85.7%. Although this paper has made some progress in monitoring and early warning, it still needs to strengthen the quantitative analysis process of monitoring indicators to achieve long-term rock burst monitoring and early warning.

Reviewer 2 Report
The work is devoted to the actual problem of forecasting mountain impacts by geophysical methods based on the registration of acoustic and electromagnetic emission signals. The problem of decoding measurement data based on a statistical approach to recognizing the state of the monitoring object is considered. the parameters informative in relation to the impact hazard of the mountain range are proposed, which is of undoubted scientific and practical interest. However, the considered monitoring gives a short-term forecast a few hours before the dynamic manifestation of mountain pressure, it would be interesting to consider the attitude to the options for a long-term forecast, which is described.
Author Response
Thank you for your letter and comments concerning our manuscript entitled “Multi-index Geophysical Monitoring and Early-warning for Coal Burst in Coalmine: A Case Study” (Manuscript Number: IJERPH-2083398). Those comments are all valuable and very helpful for revising and improving our paper, as well as the important guiding significance to our research. We have studied the comments carefully and made corrections, which we hope meet with approval. Revised portions are marked in red on the paper. The main corrections in the paper and the responses to the reviewer’s comments are as following:
The work is devoted to the actual problem of forecasting mountain impacts by geophysical methods based on the registration of acoustic and electromagnetic emission signals. The problem of decoding measurement data based on a statistical approach to recognizing the state of the monitoring object is considered. the parameters informative in relation to the impact hazard of the mountain range are proposed, which is of undoubted scientific and practical interest. However, the considered monitoring gives a short-term forecast a few hours before the dynamic manifestation of mountain pressure, it would be interesting to consider the attitude to the options for a long-term forecast, which is described.
Response: Thank you very much for your important comment. The development trend is the quantitative analysis of early warning parameters in rock burst early warning technology. This paper mainly analyzes and discusses microseismic, electromagnetic radiation and acoustic emission signals, but it needs more analysis and research on stress-related parameters such as mine pressure and ground stress. These parameters are also essential to rock burst monitoring and early warning work. Therefore, although this paper has made some progress in short-term monitoring and early warning, it is still necessary to strengthen the quantitative analysis process of monitoring indicators to achieve long-term rock burst monitoring and early warning.

Reviewer 3 Report
This manuscript to the readers of this periodical is very relevant. The article takes the mine rock burst disaster as the research background. The topic is derived from reality and has good theoretical significance and application value. However, there were some problems need be solved or modified.
1)There are grammatical and editorial issues in many parts of this paper. Many of them could cause difficulties to the readers. Such as the singular and plural nouns, misuse of et al.
2)Professional names should be unified, such as rock burst.
3)The references are a bit old, it is recommended to use more recent literature.
4)Figure 1 on the specific location of the Laohutai coal mine is not clear, should be clearly marked.
5)Why D-S evidence theory should be selected as a monitoring and early warning method, without mentioning specific reasons.
Author Response
Thank you for your letter and comments concerning our manuscript entitled “Multi-index Geophysical Monitoring and Early-warning for Coal Burst in Coalmine: A Case Study” (Manuscript Number: IJERPH-2083398). Those comments are all valuable and very helpful for revising and improving our paper, as well as the important guiding significance to our research. We have studied the comments carefully and made corrections, which we hope meet with approval. Revised portions are marked in red on the paper. The main corrections in the paper and the responses to the reviewer’s comments are as following:
1)There are grammatical and editorial issues in many parts of this paper. Many of them could cause difficulties to the readers. Such as the singular and plural nouns, misuse of et al.
Response 1: Thank you very much for your important comment. The grammatical and editorial issues have been revised. Please see the attachment.
2)Professional names should be unified, such as rock burst.
Response 2: Thank you very much for your important comment. We have unified professional names, such as rock burst and microseismic.
3)The references are a bit old, it is recommended to use more recent literature.
Response 3: Thank you very much for your important comment. The references have been revised. Please see the manuscript.
4)Figure 1 on the specific location of the Laohutai coal mine is not clear, should be clearly marked.
Response 4: Thank you very much for your important comment. We have re-marked the geographical location of the Laohutai coal mine in Figure 1. Please see the manuscript.
5)Why D-S evidence theory should be selected as a monitoring and early warning method, without mentioning specific reasons.
Response 5: Thank you very much for your important comment. Evidence theory has its unique advantages : (1) Evidence theory has a strong theoretical basis, which can deal with both the uncertainty caused by randomness and the uncertainty caused by fuzziness ; (2) Evidence theory can distinguish between unknown and uncertain ; (3) Evidence theory does not need prior probability and conditional probability density, so it is convenient for application. The advantages of D-S evidence theory are supplemented in section 3.3. Please see the manuscript.
